

# Computational prediction of some properties of 3-substituted cyclooctynes and assessment of their synthetic accessibility *via* anionic vinyl bromide precursors

Patrick Bollwerk, Tate Hendrix, Peter Ivester, Sarah N. Lynch, Gregory Maddox, Charles McCloy, Reagan Meeks, Ben Mintz, William L. Moore, Hailey M. Reddersen, Chloe Scholl, Abigail Simpson, Jacob D. Sylvie, Victoria N. Teague, Isabella K. Toole, Noah L. Wilkes and Gary W. Breton

Department of Chemistry and Biochemistry, Berry College, Mount Berry, Georgia, United States

Corresponding author
Gary W. Breton, gbreton@berry.edu

## ABSTRACT

Cyclooctyne, a strained cyclic alkyne, readily engages in "click reactions" with alkyl azides to form 1,2,3-triazoles. While there have been previous studies on the effects of substitution on the cyclooctyne framework in an effort to tune reactivity, little work has been directed towards substitution of one of the propargyl methylene groups despite the fact that such substitution is likely to have dramatic reactivity-tuning effects. We have modelled eight cyclooctyne derivatives substituted at the propargyl site using the *ωB97X-D/aug-cc-pVDZ* density functional theory (DFT) method. The energies of these derivatives relative to the unsubstituted cyclooctyne suggests greater stabilization. Furthermore, calculated global electrophilicity indexes suggest that most will have higher reactivity toward nucleophiles such as alkyl azides. Finally, we predict the feasibility of the synthesis of these derivatives *via* computational optimization of appropriately substituted vinyl anion precursors. In almost all cases there is spontaneous elimination of a bromide leaving group and formation of the desired substituted cyclooctyne. The work here suggests that propargyl-substituted cyclooctynes are promising synthetic targets that may exhibit enhanced reactivities beyond that of the unsubstituted compound.

## INTRODUCTION

The groundbreaking work in the Bertozzi group that led to the awarding of the 2022 Nobel Prize in Chemistry was based on the recognition of the enhanced reactivity of strained cyclooctynes (**1**) towards alkyl azides (**2**) that leads to the rapid formation of 1,2,3-triazoles (**3**) (see Fig. 1) (*Agard, Prescher & Bertozzi, 2004*). This variation of "click chemistry" that rapidly connects two smaller molecules to form more complex substrates, an approach pioneered by the Sharpless group (*Kolb, Finn & Sharpless, 2001*), has proven itself to be

**Figure 1** **The general reaction of cyclooctyne (1) with alkyl azides (2) to form 1, 2, 3-triazoles (3).**

remarkably robust even in living systems (*Jewett, Sletten & Bertozzi, 2010*). Since its reporting, many researchers have investigated the effect of modifying the structure of the cyclooctyne ring in an effort to tune reactivity (*Jewett, Sletten & Bertozzi, 2010*; *Debets et al., 2010*; *Kang & Kim, 2021*; *Mbua et al., 2011*; *Ning et al., 2008*; *Schuhmacher, Beile & Meier, 2013*; *Starke, Walther & Pietzsch, 2010*; *Yoshikawa, Hamada & Matsuo, 2025*; *Reese & Shaw, 1970*; *Hagendorn & Brase, 2014*). Some representative examples are provided in Fig. 2. Many of these modifications have included replacement of one or more of the methylene groups of the cyclooctyne ring system with heteroatoms or other groups. There has been particular emphasis in substitution at the 5-position relative to the alkyne group (*e.g.*, compounds **4** and **6**) as well as substitution of one of the propargylic methylene hydrogen atoms (compounds **5**) (*Debets et al., 2010*; *Kang & Kim, 2021*; *Mbua et al., 2011*; *Ning et al., 2008*; *Schuhmacher, Beile & Meier, 2013*; *Starke, Walther & Pietzsch, 2010*; *Yoshikawa, Hamada & Matsuo, 2025*; *Reese & Shaw, 1970*; *Hagendorn & Brase, 2014*). Interestingly, however, little work has been reported on the synthesis of cyclooctynyl systems with direct replacement of the entire methylene group located at the propargylic site although some (*i.e.*, $7_S$, $7_{CF2}$, $7_{CO}$) are known (see Fig. 3) (*Eaton & Stubbs, 1967*; *Madea, Slanina & Klan, 2016*; *Meier, Stavridou & Storek, 1986*; *Stavridou, Schuhmacher & Meier, 1989*). The relative lack of these derivatives is surprising since such substitution would be expected to have a much more direct "tuning" effect on the reactivity of the triple bond than substitution at distal sites whose effects would likely be attenuated by intervening saturated carbon atoms. In addition to the already-known derivatives, we were interested in the potential effects of other substituents at this site as illustrated by compounds $7_X$ in Fig. 3. Notice that the functionality of $7_{NH}$, $7_{C\equiv CH2}$, and $7_{CO}$ could allow for further derivatization if desired. Indeed, the ethylene glycol ketal of $7_{CO}$ is also known (*Kang & Kim, 2021*). One of the primary methods by which cyclooctynes have been synthesized has been *via* elimination of a leaving group (typically Br or OTs) from a suitably substituted cyclooctene precursor as illustrated in Fig. 4. The most commonly employed method has been *via* base-induced elimination of a leaving group (often using LDA) (see, for example, *Debets et al., 2010*; *Kang & Kim, 2021*; *Mbua et al., 2011*; *Ning et al., 2008*; *Madea, Slanina & Klan, 2016*). Indeed, of the three known propargylic-substituted derivatives in Fig. 3 for which we had interest in investigating, compounds $7_{CF2}$ and $7_{CO}$ had been synthesized by just such an elimination process (*Eaton & Stubbs, 1967*; *Madea, Slanina & Klan, 2016*). Additionally, fluoride-promoted

**Figure 2 Structures of some previously-studied modified cyclooctynes 4–6.**

**Figure 3 Structures of 3-substituted cyclooctynes, $7_X$, that are the subject of the current study.**

**Figure 4 Various ways in which the anionic precursor molecules 8, that ultimately lead to formation of substituted cyclooctynes, have been generated.**

elimination of a vinyl TMS group followed by elimination of a leaving group has been used for the synthesis of not only 8-membered cyclooctynes, but also for even more highly strained 6-membered cyclohexynes (*Jewett, Sletten & Bertozzi, 2010*; *Shah, Medina & Garg, 2016*; *Tlais & Danheiser, 2014*).

Previous work from our group has demonstrated that the feasibility of synthesis of strained organic molecules such as anti-Bredt bridgehead alkenes and [1.1.1] propellanes, including those with extensive heteroatom substitution, can be predicted computationally *via* optimization of the geometry of anionic precursors that contain appropriately placed leaving groups (*Breton & Ridlehoover, 2024*; *Gates et al., 2023*). If the geometry optimization of an anionic precursor leads to spontaneous elimination of the leaving group to form the desired target molecule, the product of the reaction is obviously at a lower energy on the potential energy surface. Therefore, such computational behavior is suggestive that such a reaction might also readily take place experimentally. If, however, the geometry optimization of an anionic precursor leads instead to an energy minimum structure without elimination of the leaving group, it suggests there is an energetic barrier (of unknown height) towards formation of the desired target molecule. Those precursors that spontaneously eliminate the leaving group to form the desired products are, therefore, the most promising target molecules for investing the time, energy, and resources required for a laboratory-based synthetic investigation (*Breton & Ridlehoover, 2024*; *Gates et al., 2023*). Extending our previous work, therefore, we were interested in using a similar geometry optimization protocol of substituted vinyl anion precursors such as **8** that could potentially lead to the novel substituted cyclooctynes shown in Fig. 3 whose reactivities could be substantially greater relative to those cyclooctynes already investigated. Those vinyl anions that spontaneously form the desired cyclooctynes upon optimization would be the most promising for further synthetic pursuit.

## MATERIALS AND METHODS

All calculations were performed using the software package GAMESS (Version = 30 JUN 2019, R1) (*Barca et al., 2020*) as implemented through the ChemCompute.org website (*Perri & Weber, 2014*). Geometry pre-optimizations of all the compounds studied, including the vinyl anionic precursors, were performed using the MMFF94 force field. Note that elimination was never observed to occur during the pre-optimization of the vinyl anionic precursors for any of the substrates studied. Complete geometry optimizations were then conducted at the *ωB97X-D/aug-cc-pVDZ* level of theory as it has proven robust for similar hetero-atom substituted substrates as well as strained substrates that we studied in previous works (*Breton & Ridlehoover, 2024*; *Gates et al., 2023*). We selected THF as the solvent for optimization (employing the polarizable continuum model (PCM)) (*Mennuchi, 2012*) because it is a commonly used solvent for experimental work with carbanions. Frequency calculations were carried out at all stationary points to ensure the lack of imaginary frequencies. Relevant output files are provided as Supplemental Information. To ensure that the final arrived-upon geometries represented the lowest energy conformations, conformation searches for each were conducted using the embedded protocol found in the software package of Spartan'24 v1.00 (*Spartan'24, 2025*). Global

electrophilicity values were calculated according to literature procedure (*Ben El Ayouchia et al., 2018*) on Spartan'24 v1.00 (*Spartan'24, 2025*) employing B3LYP/6-31G* optimized geometries and HOMO/LUMO energies to facilitate comparison to values previously published for several unsubstituted cycloalkynes at the same level of theory (*Domingo, Rios-Gutierrez & Perez, 2016*).

## RESULTS AND DISCUSSION

The optimized geometries (*ωB97X-D/aug-cc-pVDZ*) of cyclooctyne and the substituted cyclooctynes are provided in Fig. 5 along with some geometric measurements. The strain imparted to cyclooctynes, which contributes heavily to their enhanced reactivity, has been shown to strongly correlate with the bond angle(s) about the alkyne group (*Hamlin et al., 2019*; *Bettens et al., 2020*; *Josephson, Pezacki & Nakajima, 2025*). Significant deformation beyond the optimal ~180° is known to introduce strain into the system. For example, parent cyclooctyne **1**, with C-C≡C bond angles of 157.4° is estimated to introduce on the order of ~20 kcal/mol of ring strain (*Agard, Prescher & Bertozzi, 2004*). As can be seen in Fig. 5, substitution of the methylene group in derivatives $7_X$ leads to changes in the bond angles for all the cyclooctynes. With the exceptions of $7_{SO}$ and $7_{SO2}$, however, the bond angle changes are not particularly dramatic, and the impact on the geometries of the remaining methylene groups in the ring is minimal as can be visually discerned from the similarities of the structures in Fig. 5. In most cases (*i.e.*, $7_{CF2}$, $7_S$, $7_{SO}$, $7_{SO2}$, $7_{CCH2}$, and $7_{CO}$) the C-C≡C bond angle is relieved of more angle strain relative to the C≡C-X bond angle suggesting less of an energy penalty for deforming the C≡C-X bond angles. Only for $7_O$ and $7_{NH}$ does the C-C≡C bond angle constrict more than that of **1**, apparently to allow for expansion about the C≡C-X bond angle. For the sulfur series of compounds $7_S$, $7_{SO}$, and $7_{SO2}$ the C-C≡C bond angle becomes increasingly relaxed (162.3°, 165.4°, and 169.2°, respectively), while the C≡C-X angle becomes increasingly acute (157.1°, 149.0°, and 144.2°, respectively). Finally, note that unlike the bond angles, there is very little change in the C≡C bond lengths amongst the derivatives.

To gauge the impact that substitution has upon the energies of the substituted derivatives, we employed the isodesmic reaction illustrated in Fig. 6. Written in this manner, a positive change in energy suggests that substitution raises the energy of the system relative to that of cyclooctyne **1**, while a negative energy suggests a stabilizing effect. The results of these calculations for the various substituents are provided in Table 1. Interestingly, substitution of the methylene group of cyclooctyne with almost all the substituents (with the exception of $7_{CCH2}$) leads to stabilization of the cyclooctyne ring system relative to unsubstituted **1**. The stabilization may be partly due to the electronic effects (inductive and/or resonance) of the substituents, but the bulk of such effects should be effectively subtracted out according to how the isodesmic reaction is constructed. The stabilization, therefore, is more likely due to the greater flexibility of the ring system as a result of substitution allowing for relaxation of the C-C≡C and/or C≡C-X bond angles. Of the eight derivatives studied, the four derivatives in which the C-C≡C bond angle in particular experiences a significant reduction in angle strain (*i.e.*, greater than ~5°) relative to cyclooctyne **1** (*i.e.*, $7_{CF2}$, $7_S$, $7_{SO}$, and $7_{SO2}$) also enjoy substantially greater stabilization

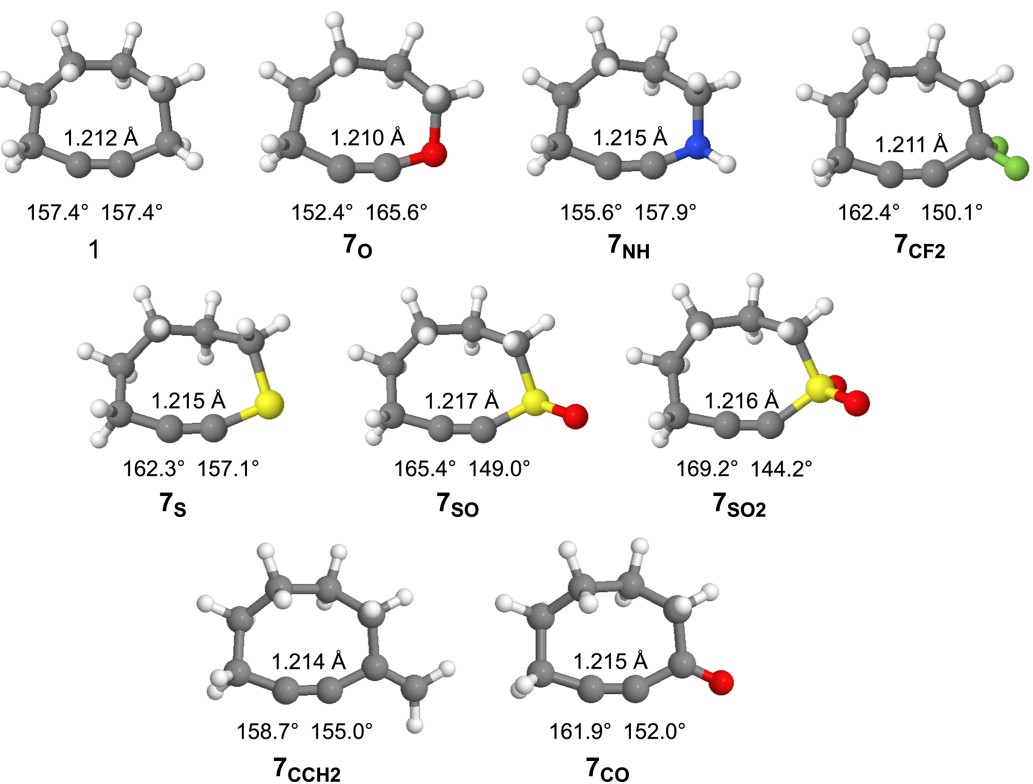

**Figure 5 Optimized geometries of cyclooctyne 1 and substituted cyclooctynes, $7_X$, with C-CC and CC-X bond angles and the CC bond lengths provided.**

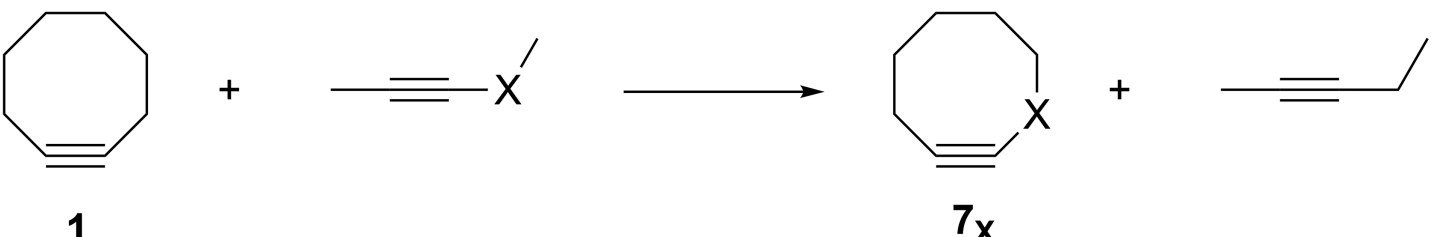

**Figure 6 Isodesmic reaction used to gauge the effect of substituents X on strain energy.**

relative to **1** (*i.e.*, > 1 kcal/mol). For $7_{NH}$, however, where the C-C≡C bond angle suffers some additional angle strain and the C≡C-X is hardly relieved of strain, there is observed lesser stabilization (−0.52 kcal/mol). Introducing the sp² hybridized carbon in $7_{CCH2}$ affords slight relaxation of the C-C≡C bond angle but somewhat constricts the C≡C-X beyond that of **1** (*i.e.*, 155.0°). Despite the small deviations in bond angles, however, the ΔE increases relative to **1** (+0.30 kcal/mol). Perhaps the increase in energy is due to the constriction of the bond angle about the sp² hybridized carbon from 116.1° in the acyclic alkyne to 110.8° in $7_{CCH2}$. Cyclooctyne $7_{CO}$ also has an sp² hybridized carbon, but it allows

**Table 1  The effect of substitution on cyclooctyne energies.**

| X | E Substituted cyclooctyne (Hartrees)[a] | E Substituted alkyne (Hartrees)[b] | ΔE (kcal/mol)[c] |
|---|---|---|---|
| $CH_2$ | −311.93809 | −195.24297 | 0.00 |
| O | −347.81162 | −231.11359 | −1.83 |
| NH | −327.96352 | −211.26757 | −0.52 |
| $CF_2$ | −510.39737 | −393.70047 | −1.12 |
| S | −670.82646 | −554.12823 | −1.95 |
| SO | −746.00358 | −629.30308 | −3.38 |
| $SO_2$ | −821.22149 | −704.51868 | −4.83 |
| $C=CH_2$ | −350.01360 | −233.31895 | +0.30 |
| CO | −385.94237 | −269.24596 | −0.70 |

**Notes:**
[a] The energy of the substituted cyclooctynes **7x** (*ωB97X-D/aug-cc-pVDZ*).
[b] The energy of the substituted acyclic alkynes (*ωB97X-D/aug-cc-pVDZ*).
[c] The change in E according to to Fig. 6.

for a greater relaxation of the C-C≡C bond angle, and slightly less of a difference in the change in bond angle about the carbonyl carbon from the acyclic alkyne to the cycloalkyne (116.1° and 111.9°, respectively). Finally, cyclooctyne $7_O$ is unique in that the C-C≡C bond angle experiences substantially greater angle strain (152.4° *vs.* 157.4° for **1**) but is still stabilized (−1.83 kcal/mol). In this case, however, the C≡C-X bond angle enjoys a greater relaxation (*i.e.*, 165.6°) than any of the other derivatives which likely compensates for the increased C-C≡C bond angle strain. Note that the greatest stabilizations are observed for the sulfur-substituted derivatives $7_S$, $7_{SO}$, and $7_{SO2}$ (−1.95, −3.38 and −4.83 kcal/mol, respectively). Thus, the stabilizing effect of the sulfur atom increases with increasing state of oxidation.

Perez has shown that the reactivity of dipolarophiles with dipoles such as alkyl azides can be predicted using the global electrophilicity index (*i.e.*, ω, in eV) (*Perez et al., 2003*) as originally conceived by *Parr, Szentpaly & Liu (1999)*. The global electrophilicity index is defined as $\omega = \mu^2/2\eta$ where $\mu = (E_{HOMO} + E_{LUMO})/2$ and $\eta = E_{LUMO} - E_{HOMO}$. Since the cyclooctynes act as the electrophilic partner in these cycloadditions (*Domingo & Acharjee, 2020*), global electrophilicity values that are higher than that of unsubstituted cyclooctyne **1** portend greater reactivity towards alkyl azides, and lower values indicate decreased reactivity. We calculated the global electrophilicity index for all the substituted cycloalkynes at the B3LYP/6-31G* level of theory for easy comparison to some previously calculated values found in the literature (*Domingo, Rios-Gutierrez & Perez, 2016*). These values are collected in Table 2, column 5. Relative to cyclooctyne **1** with ω = 0.49 eV, substitution generally increases the global electrophilicity index. Indeed, the values for $7_{CF2}$, $7_{SO}$, and $7_{CCH2}$ place them in the category of being moderate electrophiles (*i.e.*, 1.5 > ω > 0.8 eV), while those for $7_{SO2}$ and $7_{CO}$ are nearly categorized as being strong electrophiles (ω > 1.5 eV) (*Domingo, Rios-Gutierrez & Perez, 2016*). The remainder have values comparable to that of the parent cyclooctyne **1**. Only $7_{NH}$ exhibits a lower index value (0.41 eV). It's interesting to note that the global electrophilicity indices for a number

**Table 2 Calculated global electrophilicity values.**

| cmpd | X | $E_{HOMO}$ (eV)[a] | $E_{LUMO}$ (eV)[a] | $\omega$ index (eV) |
|------|-----|------|------|------|
| 7 | CH$_2$ | −6.4 | 1.0 | 0.49 |
| | O | −5.9 | 0.7 | 0.51 |
| | NH | −5.3 | 0.8 | 0.41 |
| | CF$_2$ | −7.3 | 0.0 | 0.91 |
| | S | −5.6 | 0.0 | 0.70 |
| | SO | −6.1 | −0.6 | 1.02 |
| | SO$_2$ | −7.6 | −1.1 | 1.46 |
| | C=CH$_2$ | −6.0 | −0.2 | 0.83 |
| | CO | −6.8 | −1.2 | 1.43 |
| 4 | NH | −5.5 | 0.9 | 0.41 |
| | S | −5.9 | 0.9 | 0.46 |
| | SO | −5.8 | 0.6 | 0.53 |
| 5 | OH | −6.6 | 0.6 | 0.63 |
| | NH$_2$ | −6.3 | 0.7 | 0.56 |
| | F | −6.8 | 0.6 | 0.65 |

**Note:**
[a] Calculated at B3LYP/6-31G*.

of these derivatives suggest reactivities comparable to that of the much more strained molecule cycloheptyne ($\omega$ = 0.74 eV) and even that of cyclohexyne ($\omega$ = 1.20 eV) (*Domingo & Acharjee, 2020*). For comparison, we also calculated the electrophilicity indices of previously studied cyclooctynes $4_X$ and $5_X$ (see Fig. 2). For compounds $4_X$, substitution of a methylene group at a site further than the propargylic site of compounds $7_X$ does not impart any dramatic deviation from the electrophilicity index of unsubstituted $7_{CH2}$. For example, substitution of a sulfur atom in $4_S$ leads to an electrophilicity index of 0.46 eV, while for $7_S$ the value is 0.70 eV. Substitution of one of the propargylic CH bonds in compounds $5_X$ has a more dramatic effect on the electrophilicity indices than that of $4_X$, but certainly not to the same extent as several of compounds $7_X$ (*e.g.*, compare $5_F$ [$\omega$ = 0.65 eV] to $7_{CF2}$ [$\omega$ = 0.91 eV]). Thus, as originally hypothesized, the electronic tuning effect on the cycloalkyne bond is maximized with the complete propargylic substitutions of compounds $7_X$.

Formation of substituted cyclooctynes $7_X$ from vinyl anion precursors can potentially occur from two different directions (see Fig. 7). The two regioisomeric precursors are differentiated by naming one the α-anion precursor and the other the β-anion precursor as shown in Fig. 7. The geometries of both precursors were optimized using $\omega$B97X-D/aug-cc-pVDZ to observe whether they optimized to a local minimum or spontaneously lost the Br⁻ leaving group and formed the substituted cyclooctyne. Not surprisingly, optimization of the parent vinyl anion (*i.e.*, X = CH$_2$ in Fig. 7) resulted in formation of the cyclooctyne **1** in accordance with known experimental results as shown in Fig. 8 and as movie (Video_S1) in the Supplemental Information. The optimization results of the substituted

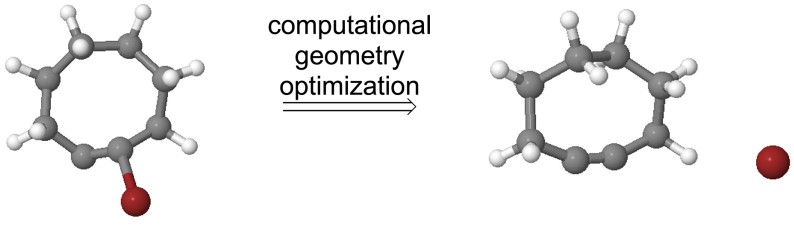

**Figure 7  Potential vinyl anion precursors to the substituted cyclooctynes.**

computational
geometry
optimization

*starting geometry*                    *optimized products*

**Figure 8  The result of computational geometry optimization of the vinyl anion precursor to cyclooctyne 1 with Br- as the leaving group (red atom).**

derivatives $7_X$ are provided in Table 3. For those precursors that did not eliminate the leaving group, a frequency calculation was performed to confirm that the resulting geometry was at an energy minimum.

As can be seen from the table, nearly all the optimizations led seamlessly to formation of the corresponding cyclooctyne *via* elimination of the leaving group. Only for one case, the α-anion precursor to $7_O$, did the elimination fail to take place. However, elimination from the regioisomeric β-anion precursor of $7_O$ was observed to take place. Using the calculated energy for the optimized α-anion precursor of $7_O$ (−2922.12254 H), ΔG for formation of the corresponding substituted cyclooctyne was calculated to be −35.6 kcal/mol. Hence, the reaction is exergonic as would be expected based on the stabilizing effects of the oxygen substituent as revealed in Table 1. It is likely, therefore, that the failure to eliminate for this precursor is because of stabilization of the negative charge at the α-position due to the inductive electron-withdrawing effects of the oxygen. This stabilization results in optimization of the anion precursor into an energy minimum with some effective barrier to elimination of the leaving group. Overall, however, these calculations suggest that substituted cyclooctynes $7_X$ could be efficiently produced *via* anion-induced elimination of vinyl bromides. Finally, we also optimized the corresponding α- and β-anion precursors to known cyclooctynes $4_X$ and $5_X$. As can be seen in Table 3, optimization of all these precursors led seamlessly to the expected cyclooctyne products. The success of this method

**Table 3  The results of computational geometry optimization of vinyl anion precursors.**

| cmpd | Substituent | Optimization yields cyclooctyne?[a] |
|---|---|---|
| 7 | $CH_2$ | Yes |
| | O-$\alpha$ | No |
| | O-$\beta$ | Yes |
| | NH-$\alpha$ | Yes |
| | NH-$\beta$ | Yes |
| | $CF_2$-$\alpha$ | Yes |
| | $CF_2$-$\beta$ | Yes |
| | S-$\alpha$ | Yes |
| | S-$\beta$ | Yes |
| | SO-$\alpha$ | Yes |
| | SO-$\beta$ | Yes |
| | $SO_2$-$\alpha$ | Yes |
| | $SO_2$-$\beta$ | Yes |
| | C=$CH_2$-$\alpha$ | Yes |
| | C=$CH_2$-$\beta$ | Yes |
| | C=O-$\alpha$ | Yes |
| | C=O-$\beta$ | Yes |
| **4** | NH-$\alpha$ | Yes |
| | NH-$\beta$ | Yes |
| | S-$\alpha$ | Yes |
| | S-$\beta$ | Yes |
| | SO-$\alpha$ | Yes |
| | SO-$\beta$ | Yes |
| **5** | OH-$\alpha$ | Yes |
| | OH-$\beta$ | Yes |
| | $NH_2$-$\alpha$ | Yes |
| | $NH_2$-$\beta$ | Yes |
| | F-$\alpha$ | Yes |
| | F-$\beta$ | Yes |

**Note:**
[a] "Yes" means that the corresponding cyclooctane was spontaneously formed during the computational optimization of the vinyl anion precursor along with loss of $Br^-$ as a leaving group. "No" means that the vinyl anion optimized as a local minimum.

in the formation of the known cyclooctyne compounds affords further confidence to the computational predictions for the formation of compounds $7_X$.

## CONCLUSIONS

Cyclooctyne **1** is known to readily engage in reactions with alkyl azides to afford 1,2,3-triazoles. We hypothesized that substitution of one of the propargylic methylene groups of **1** could be used to tune the reactivity of the cyclooctyne. Furthermore, by using the computational technique of geometry optimization of suitably substituted anionic

precursors, we could predict which cyclooctynes had the greatest promise for successful syntheses. By making use of an appropriately constructed isodesmic reaction, we have demonstrated that substitution of the propargylic methene group with the groups studied generally leads to stabilization of the cyclooctynyl system relative to the unsubstituted compound. The extent of stabilization loosely correlates with the ability of the optimized geometry to relieve angle strain at the C-C≡C bond in particular. The C≡C-X bonds appear to cost less energetically to deform beyond the preferred angle of ~180°. The tuning of reactivity was investigated by calculating the global electrophilicity index (*i.e.*, $\omega$). For most derivatives, an increase in the global electrophilicity index was observed suggesting enhanced reactivity relative to **1**. For some cases (*i.e.*, $7_{SO2}$ and $7_{CO}$) exceptionally high reactivity is predicted, even well beyond those of previously studied compounds $4_X$ and $5_X$. Finally, by computational optimization of substituted anionic precursors it was shown that all the substituted cyclooctynes of interest have great promise to be synthesized by an appropriately planned elimination reaction. The $7_{SO2}$ derivative is especially attractive as a target molecule because of its predicted stabilization relative to **1** along with its increased electrophilicity.

### Funding
Berry College provided financial support for this project. The funders had no role in study design, data collection and analysis, decision to publish, or preparation of the manuscript.

### Grant Disclosures
The following grant information was disclosed by the authors:
Berry College.

### Competing Interests
The authors declare that they have no competing interests.

### Author Contributions

- Patrick Bollwerk performed the experiments, analyzed the data, performed the computation work, authored or reviewed drafts of the article, and approved the final draft.
- Tate Hendrix performed the experiments, analyzed the data, performed the computation work, authored or reviewed drafts of the article, and approved the final draft.
- Peter Ivester performed the experiments, analyzed the data, performed the computation work, authored or reviewed drafts of the article, and approved the final draft.
- Sarah N. Lynch performed the experiments, analyzed the data, performed the computation work, authored or reviewed drafts of the article, and approved the final draft.

- Gregory Maddox performed the experiments, analyzed the data, performed the computation work, authored or reviewed drafts of the article, and approved the final draft.
- Charles McCloy performed the experiments, analyzed the data, performed the computation work, authored or reviewed drafts of the article, and approved the final draft.
- Reagan Meeks performed the experiments, analyzed the data, performed the computation work, authored or reviewed drafts of the article, and approved the final draft.
- Ben Mintz performed the experiments, analyzed the data, performed the computation work, authored or reviewed drafts of the article, and approved the final draft.
- William L. Moore performed the experiments, analyzed the data, performed the computation work, authored or reviewed drafts of the article, and approved the final draft.
- Hailey M. Reddersen performed the experiments, analyzed the data, performed the computation work, authored or reviewed drafts of the article, and approved the final draft.
- Chloe Scholl performed the experiments, analyzed the data, performed the computation work, authored or reviewed drafts of the article, and approved the final draft.
- Abigail Simpson performed the experiments, analyzed the data, performed the computation work, authored or reviewed drafts of the article, and approved the final draft.
- Jacob D. Sylvie performed the experiments, analyzed the data, performed the computation work, authored or reviewed drafts of the article, and approved the final draft.
- Victoria N. Teague performed the experiments, analyzed the data, performed the computation work, authored or reviewed drafts of the article, and approved the final draft.
- Isabella K. Toole performed the experiments, analyzed the data, performed the computation work, authored or reviewed drafts of the article, and approved the final draft.
- Noah L. Wilkes performed the experiments, analyzed the data, performed the computation work, authored or reviewed drafts of the article, and approved the final draft.
- Gary W. Breton conceived and designed the experiments, performed the experiments, analyzed the data, performed the computation work, prepared figures and/or tables, authored or reviewed drafts of the article, and approved the final draft.

## Data Availability

The unedited output files from GAMESS calculations are available in the Supplemental Files.

## Supplemental Information

Supplemental information for this article can be found online at http://dx.doi.org/10.7717/peerj-ochem.14#supplemental-information.

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
