# Peer review of "Computational prediction of some properties of 3-substituted cyclooctynes and assessment of their synthetic accessibility *via* anionic vinyl bromide precursors"

_PeerJ Organic Chemistry, doi:10.7717/peerj-ochem.14_

## Round 0.1 · original submission · Minor Revisions

Please address the following comments from reviewers in the re-submitted manuscript:

1. Explain whether multiple conformations were considered, and whether the conformers used were the lowest-energy ones.

2. Some of the structures mentioned in the introduction (for example, compound 5 in Fig. 2) are not included in the summary analysis. This will need to be explained (or the analysis could be updated).

3. The language related to synthetic feasibility must be adjusted.

4. The manuscript must be proofread once again for grammar.

**Language Note:** The review process has identified that the English language must be improved. PeerJ can provide language editing services - please contact us at [email protected] for pricing (be sure to provide your manuscript number and title). Alternatively, you should make your own arrangements to improve the language quality and provide details in your response letter. – PeerJ Staff

Reviewer 1 ·

Basic reporting

This paper described the computational modelling of eight cyclooctyne derivatives substituted at the propargyl site and compare the stabilities vs unsubstituted cyclooctyne. The authors state as main conclusion that propargyl-substituted cyclooctynes are promising synthetic targets that may exhibit
enhanced reactivities beyond that of the unsubstituted compound. Although there are some other computational studies dealing with reactivity of cyclooctynes in different reactions, as far as I could find they are not the same than the study reported in this paper.
Overall the paper is interesting but the authors should try explain better the novelties of their methodology at the end of the introduction.

Experimental design

I am not an expert in computational modelling but looking at similar works it seems to me that the methodology is appropriate and well-supported in the literature. There are in principle enough level of detail provided by the authors in order to reproduce the study. Considering all this I think this part is well described.

Validity of the findings

Once again, for someone who is not an expert in the field, it seems the main conclusions are well supported by the experimental data. One of the most important observations, which is that most anionic precursors spontaneously eliminate to form the substituted cyclooctynes supports the key hypothesis of the paper.

Additional comments

English-grammar edition could be necessary.
It would be nice that the authors provide also a critical view about the limitations that they found during the study, not only what it worked but also what it did not work during the research (there are always things that do not work in any research, and in this paper it would be nice to know it).
It would be useful to provide videos for all the compounds studied in the paper.
If possible, a final visual scheme summarizing and comparing stability and electrophilicity of different derivatives would be also useful.

Reviewer 2 ·

Basic reporting

The present paper describes a computational prediction of the important properties of cyclooctynes. Cyclooctynes are of practical interest here because they are commonly used as substrates for “click” chemistry reactions that are often used to attach probe molecules to biomolecules in living systems. Here, the authors focus specifically on studying how modification of the 3-substituted position influences important properties like flexibility and synthetic accessibility of a set of 3-substituted cyclooctynes. There are three main outputs of the paper are 1) optimized structures for all the studied cyclooctynes, 2) estimates of strained energy through isodesmic reaction calculations, 3) the global electrophilicity index, and 4) the synthetic accessibility as estimated through precursor stability.

Overall, the paper is very clearly written and logically organized. I would go as far as to say it is a pleasure to read. The figures are all relevant, high quality, well labeled and adequately described.

Experimental design

The calculations performed are well motivated by the scientific questions presented in the paper. The authors are convincing that they are attempting to fill a knowledge gap in the field with respect to the accessibility of these 3-substituted cyclooctynes.

There is sufficient information provided to make replication somewhat straightforward with one exception. Could the authors explain whether they considered multiple conformations in their calculations? I did a quick check of a few of the molecules with auto3d and found that two of them did have higher lying conformations ~3kcal/mol above the lowest energy conformation. So my main request is for the authors to explain how they found conformers and confirmed that they are the lowest energy ones.

I noticed that the authors omitted from their analysis several known examples that they discussed in their introduction. Specifically, the examples of compound 5 in Fig. 2 all fit the basic motif the authors are studying, but were not included in the analysis. Is there a reason they were omitted? Including these molecules would help strengthen the arguments of the predictiveness of the approach, since they would be additional examples of molecules that are confirmed to be accessible.

Validity of the findings

The conclusions are clearly written and mostly appropriate. I think the authors oversell the confidence of their synthetic accessibility, but since they are very clear that these are predictions, I think it’s fair to let them write as confidently as they want.

The evidence that the authors point to for the validity of their synthetic accessibility approach is less convincing than I expected based on the way it is introduced in the paper: “the feasibility of synthesis of strained organic molecules … can be confidently predicted computationally …”. The evidence in this case is two examples of reactions where this process appears to work. I’d recommend that the authors expand on this argument and provide more details of why their previous work gives them confidence in their new predictions.

---

## Round 0.2 · accepted · Accept

The updated version of the paper does address all the reviewers' concerns. I have assessed the manuscript and believe it is now ready for publication.